META-RESEARCH ARTICLE

# Gender imbalances among top-cited scientists across scientific disciplines over time through the analysis of nearly 5.8 million authors

**John P. A. Ioannidis**[1,2,3,4,5]\*, **Kevin W. Boyack**[6], **Thomas A. Collins**[7], **Jeroen Baas**[8]

**1** Department of Medicine, Stanford University, Stanford, California, United States of America, **2** Department of Epidemiology and Population Health, Stanford University, Stanford, California, United States of America, **3** Department of Biomedical Data Science, Stanford University, Stanford, California, United States of America, **4** Department of Statistics, Stanford University, Stanford, California, United States of America, **5** Meta-Research Innovation Center at Stanford, Stanford University, Stanford, California, United States of America, **6** SciTech Strategies, Inc., Albuquerque, New Mexico, United States of America, **7** Elsevier, New York, New York, United States of America, **8** Research Intelligence, Elsevier B.V., Amsterdam, the Netherlands

\* jioannid@stanford.edu

**Data Availability Statement:** The underlying data for the top 2% by age and field are provided openly in Mendeley (Collins, Thomas; Ioannidis, John; Boyack, Kevin; Baas, Jeroen (2023),

## Abstract

We evaluated how the gender composition of top-cited authors within different subfields of research has evolved over time. We considered 9,071,122 authors with at least 5 full papers in Scopus as of September 1, 2022. Using a previously validated composite citation indicator, we identified the 2% top-cited authors for each of 174 science subfields (Science-Metrix classification) in 4 separate publication age cohorts (first publication pre-1992, 1992 to 2001, 2002 to 2011, and post-2011). Using NamSor, we assigned 3,784,507 authors as men and 2,011,616 as women (for 36.1% gender assignment uncertain). Men outnumbered women 1.88-fold among all authors, decreasing from 3.93-fold to 1.36-fold over time. Men outnumbered women 3.21-fold among top-cited authors, decreasing from 6.41-fold to 2.28-fold over time. In the youngest (post-2011) cohort, 32/174 (18%) subfields had > = 50% women, 97/174 (56%) subfields had > = 30% women, and 3 subfields had = <10% women among the top-cited authors. Gender imbalances in author numbers decreased sharply over time in both high-income countries (including the United States of America) and other countries, but the latter had little improvement in gender imbalances for top-cited authors. In random samples of 100 women and 100 men from the youngest (post-2011) cohort, in-depth assessment showed that most were currently (April 2023) working in academic environments. 32 women and 44 men had some faculty appointment, but only 2 women and 2 men were full professors. Our analysis shows large heterogeneity across scientific disciplines in the amelioration of gender imbalances with more prominent imbalances persisting among top-cited authors and slow promotion pathways even for the most-cited young scientists.

"Supplementary Data for "Differential correction of gender imbalance for top-cited scientists across scientific subfields over time" ", Elsevier Data Repository, doi: 10.17632/wwykk8d48g). For the designation of "high income countries", we have used public data from the World Bank [https://datahelpdesk.worldbank.org/knowledgebase/articles/906519-world-bank-country-and-lending-groups]. All other data materials, including applied gender attributions based on NamSor, are available for scientific research purposes on ICSR Lab [https://www.icsr.net/].

**Funding:** The work of JPAI is supported by an unrestricted gift from Sue and Bob O' Donnell to Stanford University. The funders had no role in study design, data collection and analysis, decision to publish, or preparation of the manuscript.

**Competing interests:** METRICS has been funded by grants from the Laura and John Arnold Foundation (Arnold Ventures). TAC and JB are Elsevier employees and Elsevier runs Scopus which is the source of the data.

## Introduction

Gender disparities have been very prominent in science across multiple dimensions including recruitment, tenure, funding, authorship, and citation impact [1–5]. Some of these disparities may be diminishing over time, but the pace of change varies across scientific fields, settings, and countries. For example, an analysis report [6] has documented the decreasing gap between the numbers of female and male authors across science over the years, but while the gap has practically disappeared in Argentina, it continues to be very large in Japan. Moreover, in the same analysis [6] when medical subfields are examined, the inequality continues to be very strong against women in the fields of surgery and radiology and imaging, while women authors outnumber men currently in fields such as infectious diseases, fertility, and public health.

Citation impact in particular is a key coinage in the scientific academic enterprise and there is evidence that citations are misused and gamed [7]. The importance of citations both as a means of academic power as well as a marker and promoter of inequalities may be most prominent among the most-cited scientists and may also affect academic career trajectories. Gender imbalance in scientific careers may be driven be multiple complex forces [8,9], but publications and citations may be key common mediators. Differences in citation counts may reflect difference in the number of publications and/or in the citations received per publication for authors of different genders [6]. Previous analysis has shown [6] that, overall, women tend to publish fewer papers than men and that the field-adjusted number of average citations received is modestly larger when first author is a man rather than a woman.

Here, we aim to use comprehensive publication and citation data that cover all science through the Scopus database [10] in order to evaluate how gender disparities have changed over time in the cohorts of the most-cited scientists, across each of the 174 subfields of science [11]. Top-cited scientists are a select group that is the most influential across science, and any gender biases in this group are likely to have major repercussions for science at large. The available data that we have compiled allow us to investigate cohorts of scientists according to their publication age (i.e., how many years they have been active publishing scientific work). For scientist cohorts of different publication age, we identify the 2% top-cited scientists based on citation metrics that incorporate not only the number of citations received, but also information on and adjustment for co-authorship and for authorship positions among published papers.

## Methods

We used the approach we have previously applied [12–14] for generating a composite citation index and the construction of comprehensive databases including the 2% top-cited scientists in each of 174 scientific subfields, as defined by the Science-Metrix (RRID:SCR_024471) classification [11]. These subfields cover all types of science, technology, and (bio)medicine as well as scholarly disciplines on the study of humanities and social disciplines. We use summarily the terms "science," "scientific fields," and "scientists" in the paper to cover all these diverse types of scholarship, even though, strictly speaking, some of these authors may not see themselves as "scientists." Each scientist may have published papers in more than 1 subfields, but eventually he/she is classified into a single dominant subfield, the one that has the highest percentage among his/her papers. The utilized Science-Metrix classification uses allocation each journal in a single subfield, except for multidisciplinary journals where the articles may be split into multiple subfields. The primary aim of the analysis was to assess across each of the 174 scientific subfields to what extent the gender imbalance in representation of women among the top-cited scientists has been ameliorating over time. We can track the representation percentage-wise of women among the top-cited scientists with different publication ages.

We used NamSor (RRID:SCR_023935) [15], a gender-assignment software to assign gender to the Scopus (RRID:SCR_022559) author IDs. We have previously used Scopus data to assign gender to each author in projects done through the ICSR Lab, such as the European Commission She Figures report: https://ec.europa.eu/assets/rtd/shefigures2021/index.html [16] and the Elsevier Gender report https://www.elsevier.com/__data/assets/pdf_file/0011/1083971/Elsevier-gender-report-2020.pdf [17]. The NamSor application programming interface takes first/last name and country into consideration. For country, we took the first country an author publishes from as the best estimate (the country of his/her oldest published paper). Some authors may move to different countries during their career and then there is no perfect solution on which country should represent them. However, gender bias has formative influence starting from very early age, therefore assigning these authors to the country of their oldest publication is probably the most appropriate choice. Information on country of birth (which may also be different from the country of oldest publication) is not available in Scopus and extremely difficult to find for most authors. We only kept gender assignments with a confidence score >85%.

We created databases of 2% top-cited authors similar to the ones that we have created and updated on an annual basis (last updated based on September 1, 2022 data using information on over 9 million authors with at least 5 published full papers across all science) [12–14]. In brief, the scientists are ranked based on a composite science indicator that considered 6 citation metrics (total citations, h-index, co-authorship adjusted hm-index, number of citations to single-authored papers, number of citations to single and first-authored papers, and number of citations to single-, first-, or last-authored papers). The composite indicator thus takes into account not only the overall citation impact, but also co-authorship, and specifically the citation impact from papers where the author has had authorship positions that in most fields suggest greater contribution to the work.

In the current project, instead of considering all authors together, we considered in separate runs:

A. Those with first publication before 1992 (30+ years of publication age),

B. 1992 to 2001 (20 to 30 years of publication age),

C. 2002 to 2011 (10 to 20 years of publication age),

D. 2012 or later (10 or fewer years of publication age).

For each of the 4 sets, we generated the list of the 2% top-cited scientists in each of the 174 subfields with the ranking based on the composite index. We generated data both for career-long impact (all citations received at any time for all papers published at any time) and for the citation impact in the most recent calendar year, in this case 2021 (citations received in 2021 to papers published in any time). This was done twice: with and without including self-citations (as done in our previous work). Results are largely similar for career-long and single recent year impact; we report in detail here the latter (single recent year impact) and also show the main results according to the former (career-wide impact) approach. Moreover, the presented analyses consider as top-cited all scientists who are in the top-2% according to the composite index score either in the calculations excluding self-citations and/or in the calculations including self-citations. The vast majority of included scientists are in the top-2% using both calculations. We then estimated the percentage of women and men in the 4 publication age cohorts for each of the scientific subfields.

In calculating whether the proportion of women among the top-cited scientists changes over time, we excluded unclear names that cannot be assigned to a gender with >85% certainty. Gender assignment is more difficult in names from some countries than for others and it also depends on whether the full first names are available rather than just the first name initial.

We focused more on subfields that have reached a percentage of women of at least 50% (matching or outnumbering men) and of at least 30% and at what age cohorts these milestones were achieved. Comparatively, we also focused at the other end of the spectrum, subfields where the percentage of women remained below 10%.

We also calculated the relative propensity R of women versus men to be among the 2% top-cited in each subfield and publication age cohort. If there are n(w) women and n(m) men in a given subfield and given publication age cohort, and N(w) and N(m) among them are in the top-2% most-cited, then $R = \frac{N(w)n(m)}{N(m)n(w)}$.

We focus on subfields and publication age cohorts with R> = 1, i.e., where women have a larger relative representation among the top-cited scientists than their representation in the overall count of authors. Reciprocally, we also focus on subfields and publication age cohorts where R<1/3, i.e., where men have more than 3-fold relative overrepresentation among top-cited scientists than in the overall count of authors.

All these analyses were primarily performed using the global data of all scientists regardless of country. We then rerun these analyses limited to the scientists who are from high-income countries and separately for non-high-income countries and for scientists who are from the United States of America. High-income countries are those classified as such by the World Bank in 2023 (https://datahelpdesk.worldbank.org/knowledgebase/articles/906519-world-bank-country-and-lending-groups).

We also present data for the ratio of men over women among authors and, comparatively, among top-cited authors separately for each country, focusing primarily on the youngest (post-2011) cohort to see if some specific countries have ameliorated imbalances more than others. Data are presented for each country for all scientific subfields combined (country-level data per subfield have mostly very small numbers).

Finally, we assessed whether women are disadvantaged in academic recruitments and promotions even if their early work has high impact. Focusing on the youngest top-cited scientists (post-2011 cohort) who are still fairly early in their career, we selected a random sample of 100 men and 100 women. We manually checked their information online to identify how many of them are as of April 2023 in academia, industry, government, or other occupation. All retrieved sources were eligible for perusal, including, but not limited to LinkedIn, ResearchGate, Google Scholar, Frontiers Biosketch pages, and personal and CV-related pages in university and other institution websites. Among those who were located in academia, we recorded how many of them were full professors, associate professors, or assistant professors (or similar early-level faculty title). We coded different academic titles into these 3 ranks, based on perceived equivalence, e.g., for United Kingdom readers may be considered the equivalent of associate professors. The manual inspection of 100 + 100 = 200 random sample of scientists was also used to examine concurrently whether there was a substantial accuracy problem with Scopus ID assignments, i.e., whether top-cited scientists with seemingly short publication age were artifacts of older researchers with fragments of their recent research output separated from their earlier work. This artifact arises because in a few cases, the publications of an author are split in 2 or more different author ID files in Scopus that may contain papers covering different year spans; e.g., if an author who has published from 2000 until now has her publications split into a Scopus ID file that covers the papers published from 2000 until 2020 and a different Scopus ID file that covers the papers published after 2020, that second Scopus ID file will appear as if it belongs to a young author. It also allowed additional validation of the gender assignment generated by the NamSor algorithm.

This is a descriptive, exploratory analysis of a science-wide large bibliometric dataset (not pre-registered). We performed exploratory statistical testing using analysis of $2 \times 4$ tables adjusting for trend with exact test and comparison of proportions. *P*-values are two-tailed.

## Results

### Proportion of women among all authors and top-cited authors

Across a total of 9,071,122 authors with at least 5 full papers, the algorithm assigned 3,784,507 as men, 2,011,616 as women, and for 3,274,999 (36.1%), the assignment to gender was uncertain. Among the top-cited authors, 101,918 were assigned as men, 31,725 as women, and 61,672 were uncertain. Uncertain gender authors are not considered in any further calculations. Overall, men outnumbered women 1.88-fold among all authors and 3.21-fold among top-cited authors.

As shown in Table 1, there was increasing representation of women in cohorts of authors with more recent first publication year both for all authors and for top-cited authors. The ratio of men to women for all authors was 3.93 for authors who first published before 1992 and gradually decreased to 2.06 for authors first publishing in 1992 to 2001, 1.57 for authors first publishing in 2002 to 2011, and 1.36 for authors first publishing after 2011. There was larger gender inequality for top-cited authors at all age groups, with the respective ratios being 6.41, 3.48, 2.74, and 2.28. R (the ratio of ratios for top-cited and all authors) remained constant across age cohorts.

In younger age cohorts, the proportion of women among top-cited authors improved across disciplines (Fig 1). Among top-cited authors who published their first paper before

**Table 1. Proportion of top-cited women across 4 different publication age cohorts of top-cited scientists.**

| | First publication year pre-1992 | First publication year 1992–2001 | First publication year 2002–2011 | First publication year post-2011 |
|---|---|---|---|---|
| Total authors | 2,216,557 | 1,530,465 | 2,816,702 | 2,507,398 |
| Total men authors | 994,506 | 732,149 | 1,154,225 | 903,627 |
| Total women authors | 253,122 | 355,935 | 736,673 | 665,886 |
| Total uncertain authors | 968,929 | 442,381 | 925,804 | 937,885 |
| *Top-cited for single recent year impact* | | | | |
| Top-cited men authors | 28,242 | 18,082 | 30,527 | 25,067 |
| Top-cited women authors | 4,403 | 5,192 | 11,159 | 10,971 |
| Top-cited uncertain authors | 13,522 | 9,063 | 19,085 | 20,002 |
| Ratio R of top-cited women to men versus all authors women to men | 0.61 | 0.57 | 0.59 | 0.59 |
| Subfields with women > = 50% among top-cited | 5 | 18 | 21 | 32 |
| Subfields with women > = 30% among top-cited | 22 | 56 | 84 | 97 |
| Subfields with women = <10% among top-cited | 69 | 18 | 6 | 3 |
| *Top-cited for career-long impact* | | | | |
| Top-cited men authors | 29,317 | 19,211 | 31,689 | 25,046 |
| Top-cited women authors | 3,614 | 4,808 | 10,951 | 11,279 |
| Top-cited uncertain authors | 13,233 | 8,315 | 17,932 | 19,561 |
| Ratio R of top-cited women to men versus all authors women to men | 0.48 | 0.51 | 0.54 | 0.61 |
| Subfields with women > = 50% among top-cited | 4 | 15 | 15 | 29 |
| Subfields with women > = 30% among top-cited | 16 | 45 | 75 | 98 |
| Subfields with women = <10% among top-cited | 80 | 26 | 8 | 7 |

This data is derived from a breakdown of authors and their citation by gender, country, and career cohort (as well as across all cohorts) done across all subfields.

The changes in the proportion of women among all authors, and among top-cited authors across the 4 age cohorts (adjusting for trend) are statistically significant at $p < 0.001$. The changes in the proportion of subfields with women being > = 50%, > = 30%, and = <10% among the top-cited between the oldest and youngest cohort are also statistically significant at $p < 0.001$.

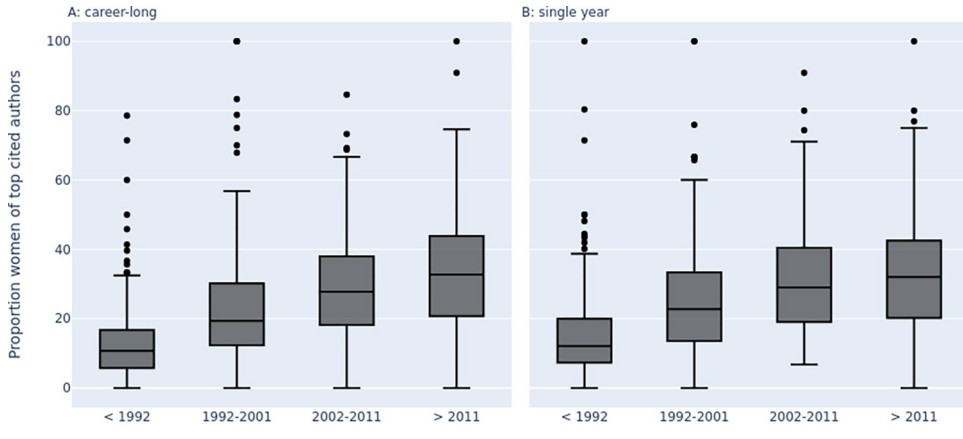

**Fig 1. Boxplots of the proportion of women among top-cited authors across the 174 scientific subfields in the 4 age cohorts.** The 2 panels show results when top-cited authors are determined according to career-long impact and when they are determined according to single recent year impact. The data underlying this figure can be found in https://doi.org/10.17632/wwykk8d48g.3.

1992, in almost half of the 174 scientific subfields ($n = 69$, 40%) women represented only 0% to 10% of the highly cited authors (i.e., ratio of men to women was $> = 9$). The number of scientific subfields with such major underrepresentation of women among the top-cited authors decreased sharply over time and among the youngest age cohort (authors with first publication after 2011), only 3 subfields had such a pattern (Table 1). In the pre-1992 age cohort, only 5 subfields had $> = 50\%$ representation of women among the top-cited authors and the number of subfields with $> = 50\%$ representation of women increased gradually to reach 32 (18%) in the post-2011 age cohort. There was a much larger, gradual increase in the number of subfields where women represented $> = 30\%$ of the top-cited authors, from 22 (13%) in the pre-1992 cohort to 97 (56%) in the post-2011 cohort (Table 1).

Table 2 shows the 32 subfields where top-cited women matched or outnumbered top-cited men in the youngest (post-2011) age cohort. As shown, in all of these 32 subfields with 1 exception (Social Science Methods), there was a larger pool of women than men authors starting their publications post-2011. The 32 subfields cover a wide variety of disciplines, with heavier concentrations in medicine, health sciences, social sciences, and cultural domains, and distinct absence of mathematical, engineering, and economics subfields. Detailed data on all 174 subfields are placed in Elsevier Data Repository, doi: 10.17632/wwykk8d48g.3.

The propensity of women to find themselves among the top-cited (after accounting for number of total available authors) exceeded the performance of men in 29/174 subfields for the pre-1992 cohort, in 33/174 subfields in the 1992 to 2001 cohort, in 17/174 subfields in the 2002 to 2011 cohort, and in only 12/174 subfields in the youngest (post-2011) cohort. At the other end of the spectrum, subfields with more than 3-fold propensity advantage for men ($R < 1/3$) decreased from 24/179 in the pre-1992 cohort, to 17/174 in the 1992 to 2001 cohort, to 14/179 in the 2002 to 2011 cohort, and 10/174 in the post-2011 cohort. The 10 subfields with $R < 1/3$ in the youngest (post-2011) cohort were Economic Theory, Econometrics, Architecture, Microscopy, Music, General Physics, Paleontology, Biophysics, Speech Language Pathology, and Mechanical Engineering.

**Table 2. Subfields with women representing > = 50% of the top-cited scientists among those with first publication post-2011.**

| Subfields | Total scientists post-2011 | Total men post-2011 | Total women post-2011 | Top-cited men post-2011 | Top-cited women post-2011 | % women among top-cited | R | Other age cohorts with top-cited women > = 50%: Pre-92/ 92-01/02-11* |
|---|---|---|---|---|---|---|---|---|
| Folklore | 86 | 22 | 42 | 0 | 1 | 100 | ND | YNY |
| Gender studies | 503 | 89 | 309 | 2 | 8 | 80 | 1.15 | YYY |
| Social work | 1,945 | 461 | 1,102 | 9 | 30 | 76.9 | 1.39 | NYY |
| Drama and theater | 181 | 42 | 90 | 1 | 3 | 75 | 1.4 | NYY |
| Nursing | 11,685 | 1,932 | 6,804 | 52 | 121 | 69.9 | 0.66 | YYY |
| Epidemiology | 2,347 | 680 | 1,114 | 14 | 31 | 68.9 | 1.35 | NNY |
| Developmental and child psychology | 5,325 | 872 | 3,687 | 33 | 70 | 68.0 | 0.50 | NYY |
| Family studies | 814 | 169 | 518 | 5 | 10 | 66.7 | 0.65 | NYY |
| Criminology | 2,890 | 937 | 1,618 | 24 | 36 | 60 | 0.87 | NNN |
| Behavioral science and comparative psychology | 2,646 | 832 | 1,400 | 19 | 28 | 59.6 | 0.91 | NNN |
| Rehabilitation | 6,779 | 2,097 | 3,283 | 47 | 69 | 59.5 | 0.94 | NYY |
| Public health | 18,050 | 4,677 | 9,830 | 135 | 179 | 57.0 | 0.63 | NNY |
| Nutrition and dietetics | 14,367 | 3,166 | 7,212 | 110 | 142 | 56.3 | 0.57 | NYY |
| Allergy | 3,749 | 1,092 | 1,784 | 27 | 33 | 55 | 0.75 | NNN |
| Pediatrics | 12,221 | 3,441 | 5,693 | 107 | 124 | 53.7 | 0.70 | NNN |
| Obstetrics and reproductive medicine | 18,304 | 4,310 | 8,666 | 137 | 156 | 53.2 | 0.57 | NNY |
| Geriatrics | 3,936 | 941 | 1,780 | 29 | 33 | 53.2 | 0.60 | NNY |
| Gerontology | 3,184 | 814 | 1,661 | 29 | 33 | 53.2 | 0.56 | NNY |
| Arthritis and rheumatology | 11,692 | 3,056 | 4,155 | 91 | 103 | 53.1 | 0.83 | NNN |
| General clinical medicine | 4,240 | 1,181 | 1,417 | 32 | 36 | 52.9 | 0.94 | NNN |
| Legal and forensic medicine | 3,071 | 1,015 | 1,148 | 26 | 29 | 52.7 | 0.99 | NNN |
| Clinical psychology | 3,846 | 953 | 2,453 | 36 | 40 | 52.6 | 0.43 | NNN |
| Psychiatry | 18,626 | 5,481 | 8,764 | 156 | 168 | 51.9 | 0.67 | NNN |
| Genetics and heredity | 8,032 | 1,981 | 3,968 | 70 | 74 | 51.4 | 0.53 | NNN |
| Veterinary sciences | 11,378 | 3,568 | 4,975 | 103 | 108 | 51.2 | 0.75 | NNN |
| Communication and media studies | 3,574 | 1,266 | 1,454 | 31 | 32 | 50.8 | 0.90 | NNN |
| Education | 23,905 | 7,507 | 11,201 | 210 | 214 | 50.5 | 0.68 | NNY |
| Endocrinology and metabolism | 19,076 | 5,285 | 8,276 | 160 | 161 | 50.2 | 0.64 | NNN |
| Social sciences methods | 642 | 253 | 238 | 6 | 6 | 50 | 1.06 | NNY |
| Art practice, history and theory | 257 | 82 | 90 | 3 | 3 | 50 | 0.91 | YYN |
| Complementary and alternative medicine | 5,059 | 1,010 | 1,242 | 26 | 26 | 50 | 0.81 | NNY |
| General psychology and cognitive Sciences | 762 | 227 | 389 | 7 | 7 | 50 | 0.58 | NNY |

R is the ratio of women to men among top-cited authors divided by the ratio of women to men among all authors. ND, not defined.

*N: No, Y: Yes (for cohorts with authors who had their first publication before 1992, in 1992–2001, in 2002–2011, and after 2011. Percentages of > = 50% for women among the top-cited had been achieved also in the pre-1992 cohort for Architecture, for the 1992–2001 cohort for Demography, Speech-Language Pathology, Industrial Relations, Music, Psychoanalysis, Development Studies, Language and Linguistics, Classics and Anthropology and in the 2002–2011 cohort for Industrial Relations and Substance Abuse; all of these subfields (with the exception of Architecture) had 25%–50% representation of women among their top-cited authors for the post-2011 age cohort. Top-cited scientists are determined based on single recent year impact.

## Analyses on high and non-high-income countries

In the overall database, there was almost double the number of authors from high-income countries (5,899,402) than from non-high-income countries (3,171,720) and uncertain gender assignment was less common in the former than in the latter (25.3% versus 56.2%). Men outnumbered women more prominently in high-income countries (2,925,898/1,481,038 = 1.98) than in other countries (858,609/530,578 = 1.62) in the number of authors. The 2 groups of countries had a similar preponderance of men over women among the top-cited authors (85,776/26,663 = 3.18 and 16,142/4,762 = 3.39, respectively).

However, when focusing on the youngest cohort (first publishing after 2011), there was an equal number of authors from high-income and other countries (1,259,314 versus 1,248,084) and both groups of countries had a similar ratio of male to female authors (576,212/415,307 = 1.36 versus 336,415/250,579 = 1.34), while the preponderance of men over women among the top-cited authors had decreased in the high-income countries, but not substantially in the other countries (17,742/8,472 = 2.09 versus 7,325/2,499 = 2.93).

As shown in Table 3, there was a gradual increase over time in the number of subfields where women represented $>$ = 50% of the top-cited authors in high-income countries (from 3% in the pre-1992 cohort to 21% in the post-2011 cohort), but this was not seen in other countries (still 6% in the post-2011 cohort). The number of subfields where women represented = <10% of the top-cited authors decreased for both high-income and other countries, but 15% of subfields in non-high-income still showed = <10% women authors even in the youngest (post-2011) cohort. Detailed data on all 174 subfields appear in Elsevier Data Repository, doi: 10.17632/wwykk8d48g.3.

## USA authors

Authors from the USA represented about 30% to 40% of the authors from high-income countries in various analyses and age cohorts and their representation of women was very similar to the overall data from all high-income countries. For example, there were overall 1,910,526 authors assigned to the USA and gender could not be assigned for 423,580 (22.2%). The ratio of male to female authors was 985,813/501,133 = 1.97 overall and 34,664/11,480 = 3.02 for top-cited authors. Among the youngest (post-2011) cohort (372,725 authors), the respective ratios

**Table 3. Number of scientific subfields with $>$ = 50% or = <10% representation of women among the top-cited scientists, in high income and non-high-income countries.**

| First publication year | Subfields with women $>$ = 50% among top-cited (high-income countries) | Subfields with women $>$ = 50% among top-cited (non-high-income countries)* | Subfields with women = <10% among top-cited (high-income countries) | Subfields with women = <10% among top-cited (non-high-income countries) |
|---|---|---|---|---|
| Pre-1992 | 5/174 (3%) | 10/128 (8%) | 72/174 (41%) | 76/128 (59%) |
| 1992–2001 | 17/174 (10%) | 27/142 (19%) | 20/174 (12%) | 47/142 (33%) |
| 2002–2011 | 22/174 (13%) | 22/154 (14%) | 10/174 (6%) | 33/154 (21%) |
| Post-2011 | 37/174 (21%) | 10/158 (6%) | 9/174 (5%) | 24/158 (15%) |

*Almost half of these occurrences (34/69) represent situations where the specific age cohort and subfield there was 1 top-cited woman and 0 or 1 top-cited man. Excluding these occurrences, the number of subfields with $>$ = 50% among the top-cited authors in non-high-income countries are 3, 15,11, and 6 in the pre-1992, 1992–2001, 2002–2011, and post-2011 cohorts, respectively.

The changes in the proportion of subfields with women being $>$ = 50% (high-income countries), = <10% (high income countries), and = <10% (non-high-income countries) among the top-cited between the oldest and youngest cohort are statistically significant at $p < 0.001$. The change in the proportion of subfields with women being $>$ = 50% (non-high-income countries) among the top-cited between the oldest and youngest cohort is not statistically significant ($p > 0.25$). Top-cited scientists are determined based on single recent year impact.

had decreased to 164,151/129,831 = 1.26 and 6,350/3,240 = 1.96. Frequency of very high or very low female representation in different subfields was similar to that found in high-income countries. Detailed data on all 174 subfields appear in Elsevier Data Repository, doi: 10.17632/wwykk8d48g.3.

## Country-level analyses

Different countries varied in the extent of imbalance for the ratio of representation of men versus women among all authors and among top-cited authors. Fig 2 presents these ratios for the youngest (post-2011) cohort for the 52 countries with more than 5,000 authors (numbers are small and make these ratios more unreliable for other countries). Results were similar when top-cited authors were determined based on career-long impact (Fig 2A) or single recent year impact (Fig 2B) and we discuss here in detail the latter. In the youngest cohort, 11 countries had fewer men than women authors (the lowest ratios were 0.63 in Thailand and 0.87 in Italy), 1 country had an equal number for both genders and 41 had more men than women (the highest ratios were 6.85 in Iraq and 4.06 in Saudi Arabia). In the youngest cohort, no countries had fewer men than women top-cited authors, but the ratio was closest to 1.00 for Italy (1.03) and Romania (1.04). Conversely, the highest ratios of men versus women top-cited authors were seen in Iraq (14.2) and Japan (9.92). India, Colombia, Pakistan, Argentina, Finland, and Japan had the worse deterioration of the gender imbalance when top-cited authors were considered rather than all authors (ratio of ratios, 4.95, 4.38, 3.70, 3.36, 2.98, and 2.92, respectively). Of these 6 countries, Argentina and Finland had more women than men authors overall, but large imbalances favoring men among top-cited authors. Detailed data on all countries and age cohorts are in Elsevier Data Repository, doi: 10.17632/wwykk8d48g.3.

## In-depth manual assessment of random samples from the youngest cohort

In the random sample of 100 authors assigned as women by the algorithm and selected from the post-2011 cohort, on close manual verification, 3 were men and 10 were authors who had actually started publishing before 2011 but their earlier publications had not been assigned by Scopus in their selected main author profile. Among the respective random sample of 100

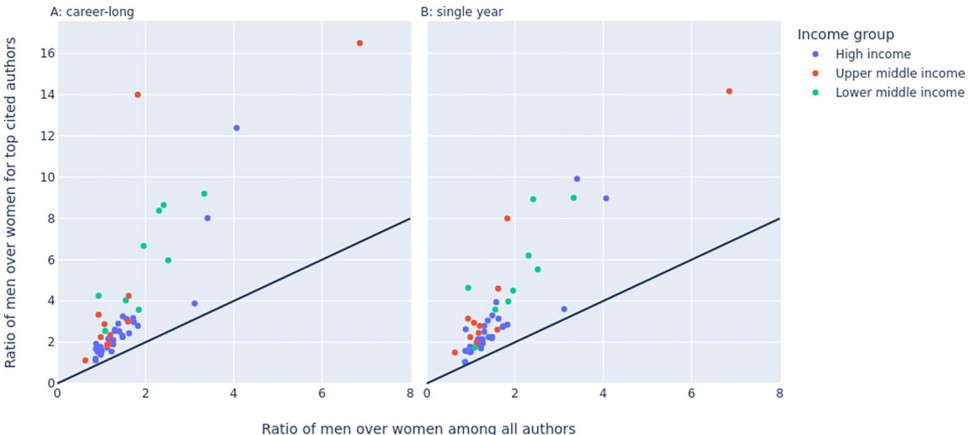

**Fig 2.** Ratio of men over women for all authors (horizontal axis) and ratio of men over women for top-cited authors (vertical axis) for the youngest age cohort (authors who started publishing after 2011) for the 52 countries that had more than 5,000 authors in that cohort. Countries with fewer authors have too few top-cited authors and the gender ratio for top-cited authors would be driven by very small numbers. The 2 panels show results when top-cited authors are determined according to career-long impact and when they are determined according to single recent year impact. The data underlying this figure can be found in https://doi.org/10.17632/wwykk8d48g.3.

**Table 4. In depth analysis of current (April 2023) occupation of random samples of men and women from the youngest (post-2011) cohort.**

| Current occupation (as of April 2023) | Women (*n* = 87)* | Men (*n* = 91)* |
|---|---|---|
| Academia | 59 | 69 |
| Full professor | 2 | 2 |
| Associate professor | 11 | 13 |
| Assistant professor of similar** | 19 | 29 |
| Other (training, staff) or unclear | 27 | 25 |
| Government | 9 | 3 |
| Other research institute (non-industry) | 5 | 2 |
| Industry | 7 | 13 |
| Clinical*** | 7 | 4 |

Data on current occupation and time onset of publication records were compiled perusing online searches with the name of the scientist and examining LinkedIn, Google Scholar, ResearchGate, Frontiers, Scopus, and any other relevant data that appeared in these searches. The most recent position was recorded, but it cannot be certain that this information online was entirely up-to-date.

*Of 100 randomly selected women, 13 were artifacts and the same applied to 9 men (see text for details).

**Lecturer, senior lecturer, instructor, or other titles that presumably are in the same level

***Clinicians with academic titles were assigned to the academic group.

authors assigned as men, 9 had similarly started published in earlier years. Thus, 87 verified eligible women and 91 verified eligible men had their work histories evaluated in-depth for their current occupations (Table 4). There were some possible trends for more women being engaged in government positions than men and more men than women working in the industry, but the difference could have been due to chance. The majority in both gender samples were in an academic environment; among them, slightly more than half already had some academic title and this tended to be more common for men than for women (44/69 (64%) versus 32/59 (54%)), but the difference was not statistically significant (*p* = 0.29). Only 4 had full professor appointments (2 women, 2 men; all 4 in non-high-income countries).

## Discussion

Our evaluation of a comprehensive science bibliometric database with over 9 million authors who have published at least 5 full papers shows that there have been substantial corrections of the gender imbalance in the scientific workforce over time. However, these corrections are still lagging behind in many scientific subfields and vary extensively across countries. Moreover, while the difference between the number of male and female authors has overall become modest (about 1.3-fold across all scientific authors), the difference in the number of top-cited authors between the 2 genders remains much higher. The overall imbalance in this regard is about 2-fold in high-income countries (and also in the USA specifically) and 3-fold in other countries. There is currently very large heterogeneity across scientific subfields and countries in the presence and prominence of gender imbalances. In the youngest cohort of scientists (those who started publishing after 2011) in almost 1 in 5 subfields, women match or outnumber men among the ranks of its top-cited scientists. However, in almost all of these subfields this largely reflects the fact that more women than men work and publish in them. Conversely, even in the youngest cohort of scientists, in 44% of the scientific subfields women represent less than 30% of the top-cited authors. Finally, most of the youngest top-cited scientists are still working in academic environments and there was a trend for more men than women to have

positions in the typical academic ladder (assistant, associate, or full professor). Nevertheless, very few have reached full professor appointments.

We also examined relative propensity metrics that correct the ratio of top-cited authors by considering also the ratio of all authors who are women versus men. We noted that over time, there were fewer subfields where women had a competitive advantage against men to find themselves among the top-cited authors once they started publishing in a given subfield. Concurrently, the number of subfields where men had a large competitive advantage to find themselves among the top-cited authors once they started publishing in a given subfield also diminished over time.

The scientific workforce globally is changing. There is a rapid advent of massive research productivity in research and scientific publications in some non-high-income countries like China [18–20], often with financial incentives that have attracted criticism [18]. Therefore, the share of authors from non-high-income countries has increased sharply. Among the youngest authors, those with a decade or less of Scopus-indexed publication history, half of them come from non-high-income countries. In these countries, gender imbalances in the ability to reach the top-cited group remain much stronger than in high-income countries. This poses an extra challenge to achieving equity in these countries, where research is often performed under suboptimal circumstances. Previous work has shown that the gender gap in science, technology, and medicine fields is smaller in countries where women are more likely to major in those fields [21]. Moreover, there are local factors and barriers in each low- and middle-income country that shape the characteristics of its workforce and the extent of distortion from gender bias [22].

Even high-income countries, including the USA, have much room to optimize equity. In the USA, it has been documented that women are less likely to be included as authors especially in highly cited papers [23]. Japan has 10-fold more top-cited men than women even in the youngest cohort, probably a reflection of long-lasting traditions [24], despite efforts to improve perceived norms [25]. Other countries such as Argentina and Finland have strong gender imbalances in top-cited authors, even though they have managed to extinguish imbalances in the overall number of authors.

Much attention needs to be given to the younger generations of scientists to promote equity and optimize their path in research. Our in-depth analysis of a random sample of scientists who have been publishing Scopus-indexed papers for a decade or less shows a trend for more men than women to have entered the academic ladder, although the samples were quite small and the difference could be due to chance. We should caution that information available online on academic ranks may not be always up-to-date or complete, but any missingness is probably not affected by gender. A worrisome observation is that very few of these early over-achievers have reached professor-level appointments. None of the 4 full professors in this sample came from a high-income country and we cannot exclude the possibility that the 4 full professors had also started publishing earlier than 2012 in journals that are not indexed in Scopus (e.g., local journals) and thus may have longer publication ages. In most academic environments and in most countries, progress through the academic ranks takes a painfully long time and funding independence is typically reached in the mid-40s [26]. Funding disparities also continue to exist and they may fuel career choices and advancement [27]. The current situation may be eroding independent creativity and needs to be challenged [28]. In the past, the time to get a doctoral degree was shorter and scientists could become faculty and even full professors very soon after obtaining their doctoral degree. Lengthy graduate studies and multiple postdoctoral experiences are currently far more common before reaching independence [29–31]. Very talented individuals who show clear early promise may need to be promoted much

faster. Perhaps overall research originality and creativity gets promoted if young talented scientists are given more support and confidence in tenure.

We considered all authorship positions in counting number of scientists of different genders. However, the calculation of the composite citation indicator that is used to identify the top-cited scientists gives a lot of weight to single, first, and last authorships, as opposed to middle authorship. It is possible that in some cases, women may be more likely to be listed as secondary or supporting authors rather than first or senior authors (or not be listed at all) [23] and this can impact their visibility and recognition in the academic community. Thus, multiple forces may converge towards diminishing the chances of women becoming top-cited.

Our work has some limitations. First, for over a third of the authors, gender assignment was uncertain and these people had to be excluded from further analyses. This level of uncertainty is unavoidable with any automated gender assignment tool. Uncertain gender was modestly less common among the top-cited scientists and in those from high-income countries, but it was very high in authors from non-high-income countries. While there is no reason to believe that the representation of men versus women would be different in the uncertain gender group, we cannot exclude the possibility for some imbalance, e.g., if women are more likely to use only initials rather than full first name. One should therefore be cautious about the uncertainty that these excluded authors induce in the main calculations of gender ratios. Moreover, even with a >85% certainty selection threshold, some people will be assigned to the wrong gender by NamSor. This wrong assignment happened nevertheless in only 6 of 200 randomly selected authors examined in depth. It is possible that the risk of mistaken gender assignment varies across fields and it may be more likely to reflect women being assigned to male gender than the opposite. If so, this may cause some underestimation of the percentage of women in some countries. Second, Scopus is a comprehensive database, but some types of publications, e.g., books and some specific journals may not be represented properly [10]. This may affect the validity of the ranking in some scientific subfields (which specific individuals are included in the top-cited), but it is less likely to affect gender ratios. Third, we used a previously extensively validated methodology for identifying the top-cited authors. However, as we have described before in detail [12–14], all citation metrics and calculations, including ours, have deficiencies and inaccuracies [32]. Moreover, citation impact (in whatever form it is calculated) should not be construed as a perfect surrogate of research quality or real-world impact. Nevertheless, our approach offers a reproducible way to identify authors with the highest citation metrics. Imbalances between genders may vary in degree across different metrics or aspects of work achievement. Finally, we could only look at the distinction between men and women, a binary classification that does not consider self-perceptions of gender which go beyond binary options. This is a known unavoidable limitation of any algorithm that tries to assign gender based on names' and countries' information.

Allowing for these caveats, this large-scale analysis offers insights for past and current status of gender imbalances in scientific productivity and top citation impact and may be used for future planning and evaluation. Similar data may also be used for benchmarking within single countries and institutions. The persisting large imbalances in several scientific disciplines need more study to understand their causes. One may also learn a lot from disciplines where women have matched or even outperformed men in productivity and citation impact. The counterfactual of ideal equity may not represent a situation where men and women have equal representation among the top-cited scientists in each and every subfield. Nevertheless, the big composite picture suggests that there is still substantial room for further correction of imbalances.

## Author Contributions

**Conceptualization:** John P. A. Ioannidis, Kevin W. Boyack, Thomas A. Collins, Jeroen Baas.

**Data curation:** Thomas A. Collins, Jeroen Baas.

**Formal analysis:** John P. A. Ioannidis, Thomas A. Collins.

**Investigation:** John P. A. Ioannidis.

**Methodology:** John P. A. Ioannidis, Kevin W. Boyack, Thomas A. Collins, Jeroen Baas.

**Writing – original draft:** John P. A. Ioannidis.

**Writing – review & editing:** John P. A. Ioannidis, Kevin W. Boyack, Thomas A. Collins, Jeroen Baas.

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
