## [Editor Report · Decision Letter 0]

3 Jul 2023

Dear John, 

Thank you for submitting your manuscript entitled "Differential correction of gender imbalance for top-cited scientists across scientific subfields over time" for consideration as a Meta-Research Article by PLOS Biology.

Your manuscript has now been evaluated by the PLOS Biology editorial staff, as well as by an academic editor with relevant expertise, and I'm writing to let you know that we would like to send your submission out for external peer review.

Once your full submission is complete, your paper will undergo a series of checks in preparation for peer review. After your manuscript has passed the checks it will be sent out for review. To provide the metadata for your submission, please Login to Editorial Manager (https://www.editorialmanager.com/pbiology) within two working days, i.e. by Jul 05 2023 11:59PM.

Best wishes,

Roli

Roland Roberts, PhD

Senior Editor

PLOS Biology

rroberts@plos.org

---

## [Decision Letter · Decision Letter 1]

15 Aug 2023

Dear John,

Thank you for your patience while your manuscript "Differential correction of gender imbalance for top-cited scientists across scientific subfields over time" was peer-reviewed at PLOS Biology. It has now been evaluated by the PLOS Biology editors, an Academic Editor with relevant expertise, and by three independent reviewers. 

In light of the reviews, which you will find at the end of this email, we would like to invite you to revise the work to thoroughly address the reviewers' reports.

You'll see that all three reviewers are broadly positive about your study, but each raises a number of concerns that will need to be addressed before further consideration. Reviewer #1 asks if the study was preregistered, notes some short-cut citations, wants you to report some additional data, asks about the universality of some academic posts, and wants more detail on how authorship position was considered. S/he also makes some recommendations for improved dataviz and suggests some additional analyses. Reviewer #2 wants more rigorous formal stats to support your assertions of differences between fields and through time; s/he has some further requests for clarifications and extra analyses. Reviewer #3 is somewhat more guarded; s/he wants you to rectify a mismatch between the claims and the choice of academic subfields, recommends pooling small subfields to make numbers of researchers in each more comparable, questions the robustness of your gender classification tools (especially if it has subfield-specific problems), and challenges the relevance of your “country” analysis.

IMPORTANT: I discussed the reviews with the Academic Editor, and they sent me the following comments, somewhat edited, which you may find helpful (and which you should address):

"I agree with most of the reviewers comments, each brings up important points and none of them are unreasonable.

"I see where the authors are coming from, no, not everything is going to need to be hypothesis tested. However, as reviewer #2 points out, descriptive stats are important for making some of the points that authors would like to make. Post hoc tests are typically used for these sorts of "additional analysis" ensuring an overall lower significance rate. Of course the n's here are quite large so I would assume that no matter which analysis is done, it will be significant so I can certainly see why the authors chose this route.

"I also agree with the reviewers about the https://namsor.app/. Much like many of our analyses, the presence of a black box component (either a commercial app or an antibody kit) needs to be treated carefully. The authors simply need to address this issue, it is not disqualifying, but should be treated more carefully.

"I would also ensure that all RRIDs were added to the manuscript. There are RRIDs for software tools and databases that are appropriate here and are not included."

Given the extent of revision needed, we cannot make a decision about publication until we have seen the revised manuscript and your response to the reviewers' comments. Your revised manuscript is likely to be sent for further evaluation by all or a subset of the reviewers.

**IMPORTANT - SUBMITTING YOUR REVISION**

*Re-submission Checklist*

*Published Peer Review*

*PLOS Data Policy*

*Blot and Gel Data Policy*

Sincerely,

Roli

Roland Roberts, PhD

Senior Editor

PLOS Biology

rroberts@plos.org

REVIEWERS' COMMENTS:

Reviewer #1:

The article "Differential correction of gender imbalance for top-cited scientists across scientific subfields over time" is very well written and an important contribution in the field of gender imbalances in the scientific community. However, there is a notable omission in the presentation of results, which could enhance the overall impact of this study. There are a few concerns that need to be addressed before publishing in PLOS Biology. 

Methods: 

* Has the study been preregistered? If so please provide the link and/or DOI. 

* Particularly the methods section contains quite some short cut citations. Linking to the previous content is fine, however, few more details would be helpful for the reader to better assess the limitations of the study (Taking shortcuts: Great for travel, but not for reproducible methods sections, bioRxiv 2022.08.08.503174; doi: https://doi.org/10.1101/2022.08.08.503174)

* The authors choose to not report the results from the career-long impact; even though they are largely similar, it would be nice to either add this information as supplement (and refer to it in the man text) or even add this data in the main text in an infographic or else. Incorporating this additional data could enrich the comprehensiveness of the study and provide readers with a more comprehensive perspective on the topic.

* Full professor versus associate or assistant professor, are those universal positions one can hold? E.g., in Germany they have the concept of habilitation and junior Prof, but the requirements are different from other countries. Thus, do you consider someone early in their career into one of these categories or rather categorize under "other" if they are in the process of habilitation -I can imagine that this information is not easily accessible via LinkedIn or else.. 

* Did you consider all authorships irrespective of the position (first, last, middle)? The order of authors listed on a scientific publication can carry significant weight and influence an author's reputation and career prospects. Women, in some cases, may be more likely to be listed as secondary or supporting authors rather than first or senior authors, which can impact their visibility and recognition in the academic community. This is not clear in the manuscript and results might be different when considering the position.

Visualization(s):

* The manuscript would highly benefit from visualizations e.g., the overall number of male versus female authors and the proportion of highly cited researcher in each group. Some of the information that is currently presented in the tables 1- 4 could easily be visualized for a better understanding. 

* A Flow chart including the number of excluded data based on unclear gender assignments or else would be helpful, especially for the inclusion or exclusion of countries as only countries with > 5000 authors are considered in your descriptive analysis. Visualizing the random sample and the respective exclusion in the flow chart would be helpful. 

* Figure 1: It would be of interest to visually include the different countries into the figure, e.g., by including a color-code for high-income vs non-high income; the 53 countries could also be provided as a list in the supplement (including whether they were considered high- or non-high-income. 

Supplementary Information:

* The authors do link to the Elsevier Data Repository; however, it would be helpful for readers to provide a list of content and refer to any supplementary information within the main text. I think the repository contains valuable information and should be better assessable (consider curation and FAIR data principles). 

* In line with the previous comment: there are few parts where data is not shown, this could be done via the supplement and linking to it. As an example: "Frequency of very high or very low female representation in different subfields was similar to that found in high income countries." Why is that data not shown and why making the separation and focus on the USA (apart from the fact that the authors are based in the states)? I would rather recommend showing the overall data for high income and then include a subsection with a focus on the US, rather than doing it the other way around. If there are any restrictions though, please mention this in the limitation section. 

Results/ Discussion: 

* How was the country-level analysis performed if an author changed their affiliation over time e.g., from non-high income to high income? In the method section the authors state that they rerun the analysis for scientists from country X or Y. Could you please clarify how the assignment was done, maybe I missed this. 

* In the random sample used for the manual analysis the authors stated in the methods section that they would describe whether top-cited scientists in the youngest cohort (publication age) were artifacts of older researchers, this is a very important point and I feel it is lacking in the results and discussion section. In general, the question relates to the position of an author within a paper -if a highly cited female first author is accompanied by a male senior author. I realize that this might be beyond the scope of this publication, however, a critical discussion regarding the diversity in the author list of a given publication would be highly appreciated.

* Considering low-income or middle-income countries and the current publishing system, how large is the proportion of (highly-cited) authors in general considering high APCs? I think a brief overview of additional barriers (even if they cannot be explored in detail here) should be included in the discussion/ limitation section.

* In that context, funding disparities in high versus non-high-income countries as well as geo-political and social structures, traditions and cultural differences might play a major role and should be discussed in more detail. In that line, the gender gap (in STEM) is smaller in countries where women are more likely to major in those fields (Aldén and Neuman, Culture and the gender gap in choice of major: An analysis using sibling comparisons, Journal of Economic Behavior & Organization, 2022.). The manuscript would highly benefit from a discussion that highlights common issues in gender inequality countries that do not belong to the global north (Rose and Hardi; "With Education You Can Face Every Struggle": Gendered Higher Education in Iraq and Iraqi Kurdistan - Part Three: The Gender Problem). Highlighting these potential (additional) causes for the existing imbalance can help to raise awareness and should not be detangled from the inequity in authorship and career prospects. 

Reviewer #2:

This study performs a large-scale quantitative analysis of citation rates for authors of different gender in across different scientific subfields and at different stages of their independent research career. The analysis was applied to a dataset previously generated by this group, using a previously developed citation metric. New to this study, the authors used an automated gender identification algorithm to label authors in their dataset and compare citation rates based on these labels. They report an overall increase in representation of women researchers over time and an increase in the relative representation of women in the group of top-2% cited researchers. However, the degree of representation by women in these groups varies substantially across fields.

This study addresses an important, challenging question with a new and valuable dataset. However, it is confusing that the authors use no statistical tests to substantiate their results. In a broad sense, the conclusion that women are generally underrepresented in research but the degree of their representation has been increasing over time is not surprising. If the authors had made claims that did not align with a confirmation bias, these results would be rejected without statistical tests supporting the claims. In particular, statements about differences between fields and changes over time should be qualified or backed up with statistical tests.

MAJOR CONCERNS

1. L. 173-74. "We avoided formal statistical testing…" The logic of this statement is not clear. Statistical tests can be used to assess results in exploratory studies. If claims are to be made about numerical differences, it seems quite reasonable to back up those claims with a significance test. If the goal here is simply to publish the expanded dataset with gender labels, that may be acceptable on its own. In this case, however, claims about numerical differences between groups should be qualified as anecdotal and not statistically supported. One might argue that the results are so obvious that no statistics are needed. But if that's the case, why bother with a numerical analysis in the first place? Alternatively, it seems like all the tables (1-3, at least) could benefit from fairly straightforward statistical tests. More specific points about these tables are below.

2. L. 269-281. "close manual verification" It is unclear what to take away from this section. At face value, the small subset shows no difference between gender groups. There is the observation that only a small fraction of authors sampled have independent faculty jobs, but is this surprising, given that most authors would complete some years of graduate school and often a postdoc before obtaining a faculty position? Can the authors make a comparison that shows time-to-independence is longer than for earlier generations of researchers? Some additional analysis is required to substantiate the claims made here.

LESSER CONCERNS

L. 179 "top-cited" Please state how this group is defined. The information is provided in the Methods, but it takes a certain amount of digging. It appears that top-cited means the subset of each group with the highest composite citation index (c-index) for papers published in the last years? Or citations in the last year? And ranked both with self-citations and without? 

L. 183. Table 1. It would be helpful if the authors could provide percentages and R scores for the different groups. It is true that they can be computed from the raw numbers, but a reader would appreciated it if were reported in the table. A figure would be even better, if the authors are able to do it. 

L. 201. A similar request applies to Table 2.

L. 232. Table 3 seems to contain the key results of the study. It's clear that gender ratios are on average approaching 50%. But is the process accelerating? Slowing down? Are there significant differences or not between wealthy and less wealthy countries? The data as presented aren't convincing. 

L. 252-267. Figure 1. This figure it hard to interpret. Please add a line of unity slope to show where values on the y axis are greater than the x axis. Consider also plotting on log-log scale, as the small values are difficult to discriminate.

L. 257 "9.87 in Italy" Typo? 9.87 seems very high.

Reviewer #3:

Summary

This study evaluates how the share of female authors among the top-cited 2% in Scopus in each of 174 research subfields has changed over time. Gender of authors is determined using the NamSor database. The authors find substantial heterogeneity across disciplines and subfields in women researchers' shares and the extent to which gender disparities in the top-cited 2% has closed over four publication-age cohorts: pre-1992, 1992-2001, 2002-2011, and 2012 or later. Additional analyses consider within-field heterogeneity in these disparities across countries. Finally, the authors selected a random sample of 200 researchers in the post-2011 cohort for additional manual web research (e.g., LinkedIn, Google Scholar, university websites, etc.) to evaluate differences by gender in academic rank and employment sector.

This paper asks an interesting question and presents plausible descriptive results. However, there are several aspects that need further clarification before publication.

Major points:

1. The description of the 174 fields as "scientific" (and the authors as "scientists") throughout the paper is confusing, as readers might reasonably presume the subjects of analysis are all in the natural sciences or at least in STEM fields. As Table 2 shows, this is not the case: subfields also include humanities, family and gender studies, and so on. I would suggest either limiting the subfields discussed to natural sciences (including health sciences) for better comparability, or changing the presentation and discussion of the paper to better reflect this.

2. As Table 2 also shows, the total number of published researchers varies tremendously across fields. Weighting subfields like Folklore (86 researchers post-2011), Drama & Theater (181), Art History (257), and Gender Studies (503) equally with subfields Public Health (18,050), Psychiatry (18,626), and Education (23,905) can yield misleading conclusions about the extent to which the broader research enterprise is approaching gender balance. Moreover, I also downloaded the full classification subfield list for Science-Metrix, and while e.g. medical topics appear to be reasonably disaggregated (allergy, pediatrics, ob/gyn, endocrinology, etc.), others (e.g., economics) are not. Different citation norms, publication frequencies and peer-review timelines across true subfields of a discipline can lead to differences in citations. At a minimum, I would recommend combining similar small fields so that the number of unique researchers is more similar across "subfields".

3. For fields with very detailed subfields, is it possible that a given highly cited researcher is showing up in multiple subfields' top 2% lists? I don't know about Scopus and Science-Metrix article classifications, but Scimago's similar system classifies many journals in more than one subfield. I also don't see, though perhaps I missed it, whether the subfield classification of authors is based on the journals in which they published or article-level subfield tags. If the former, I would wonder how high-impact scientific journals like Science and Nature are categorized.

4. My own small-N of NamSor suggests it may systematically misclassify contemporary female names as male, and performs very poorly with Vietnamese names. I am in a male-dominated field, and entered the names of each of my 32 students this fall as a test case. Of these, all male students were correctly classified, with probabilities 95% or higher. Among the 14 women, fully half would either be excluded from this study or misclassified as male. Four of the women are Vietnamese, and of those, 3 would be excluded due to "unclear" gender using the 85% cutoff. Half of the 8 native-US women (all with Anglo / Western European names) were misclassified as men, with probabilities 86%, 93%, 95%, and 99%. Now consider the following thought experiment: what if parents who give female children gender-neutral or historically-male-typed names raise children who are more likely to disregard or actively challenge gender norms about degree fields? This paper's method would then systematically undercount the share of women researchers in male-dominated fields.

5. It's not clear what the country analysis is meant to portray, if authors' country is set based on the first observed publication. For many natural sciences fields, this likely represents the location of the author's graduate program or postdoc, which (in some fields more than others) might not correspond to their national origin or where they subsequently work. At minimum, more care is needed in discussing the meaning of these results. It could also be interesting to evaluate whether authors who publish in different countries over their careers have higher citations (e.g. due to networking) vs those who remain in the same country.

Minor points:

* The pre-1992 cohort is likely to have more citations for men because the window for possible citations can be arbitrarily long, and men dominated most fields before 1980. I suggest excluding this cohort, or using a similar decade cutoff as for other cohorts, for better comparability.

* On p. 4, "there is evidence that citations are misused and gamed". I'm sympathetic to this point, but the authors need to clarify what "gamed" means in this context, and if it's important, how/whether we can detect differences by gender in "gaming" citations.

---

## [Editor Report · Decision Letter 2]

29 Sep 2023

Dear John,

Thank you for your patience while we considered your revised manuscript "Differential correction of gender imbalance for top-cited scientists across scientific subfields over time" for publication as a Meta-Research Article at PLOS Biology. This revised version of your manuscript has been evaluated by the PLOS Biology editors and the Academic Editor.

Based on our Academic Editor's assessment of your revision, we are likely to accept this manuscript for publication, provided you satisfactorily address the following data and other policy-related requests:

IMPORTANT - please attend to the following:

a) Please change your title to make it more accessible and to include some idea of the impressive scale of your study. We suggest the following selection: "Analysis of nearly 5.8 million authors reveals differential correction of gender imbalance for top-cited scientists across scientific subfields over time" or "Analysis of nearly 5.8 million authors reveals large heterogeneity across scientific disciplines in the amelioration of gender imbalances among top-cited scientists over time" or (possibly the snappiest) "Gender imbalances among top-cited scientists across scientific disciplines over time through the analysis of nearly 5.8 million authors" 

b) The Academic Editor asked me to re-iterate his/her request from the previous round about RRIDs: "The only thing they did not address that was simple and practical was the addition of RRIDs, which I am assuming they simply did not understand because they did not read the instructions to authors. https://journals.plos.org/plosbiology/s/materials-software-and-code-sharing The authors used 3 major tools. These three are commercial tools therefore there are no good scholarly papers and there will certainly not be any public github repository that will point to the code. Therefore the addition of a persistent unique identifier, the RRID, is the only way to mark which tools were used when the tools are discontinued at some future point (statistically speaking 75% of tools would likely be around for two years after publication). NamSor Version or date of access (RRID:SCR_023935) Science-Metrix Version or date of access (RRID:SCR_024471) Scopus Version or date of access (RRID:SCR_022559) Please ask the authors to include these RRIDs to the methods section with any additional version or date of access information."

c) We note that you have deposited the underlying in the Elsevier Data Repository, V1, doi: 10.17632/wwykk8d48g.1 - many thanks for doing so. However, I note that the associated licence is CC BY NC. I consulted our data team, and they tell me that the data underlying a data cannot have a more restrictive licence than our CC BY one. Please could you therefore (preferably) switch the Elsevier licence to CC BY; if this is not possible, please lodge a copy of the data in (e.g.) Zenodo and provide the Zenodo URL/DOI in the paper's Data Availability Statement and Figure legends (see next point).

d) Please cite the location of the data clearly in both Figure legends, e.g. “The data underlying this Figure can be found in https://doi.org/10.17632/wwykk8d48g.1” or “The data underlying this Figure can be found in https://doi.org/10.5281/zenodo.XXXXX”

e) Please make any custom code available, either as a supplementary file or as part of your data deposition.

We expect to receive your revised manuscript within two weeks. 

*Published Peer Review History*

*Press*

Sincerely,

Roli

Roland Roberts, PhD

Senior Editor,

rroberts@plos.org,

PLOS Biology

CODE POLICY

Per journal policy, as the code that you have generated is important to support the conclusions of your manuscript, we require that you make it available without restrictions upon publication. Please ensure that the code is sufficiently well documented and reusable, and that your Data Statement in the Editorial Manager submission system accurately describes where your code can be found.

DATA NOT SHOWN?

---

## [Editor Report · Decision Letter 3]

16 Oct 2023

Dear John,

Thank you for the submission of your revised Meta-Research Article "Gender imbalances among top-cited scientists across scientific disciplines over time through the analysis of nearly 5.8 million authors" for publication in PLOS Biology. On behalf of my colleagues and the Academic Editor, Anita Bandrowski, I'm pleased to say that we can in principle accept your manuscript for publication, provided you address any remaining formatting and reporting issues. These will be detailed in an email you should receive within 2-3 business days from our colleagues in the journal operations team; no action is required from you until then. Please note that we will not be able to formally accept your manuscript and schedule it for publication until you have completed any requested changes.

Sincerely, 

Roli

Senior Editor

PLOS Biology

rroberts@plos.org